# Autodyne Sensor Signals with Amplitude-Frequency Modulation of Radiation

**DOI:** 10.3390/s20247077

**Published:** 2020-12-10

**Authors:** Vladislav Noskov, Rinat Galeev, Evgeniy Bogatyrev, Kirill Ignatkov, Kirill Shaidurov

**Affiliations:** 1Engineering School of Information Technologies, Telecommunications and Control Systems, Ural Federal University, 620002 Yekaterinburg, Russia; v.y.noskov@urfu.ru (V.N.); k.d.shaidurov@urfu.ru (K.S.); 2Joint Stock Company Scientific Production Enterprise «Radiosviaz», 660021 Krasnoyarsk, Russia; info@krtz.su (R.G.); bogatblr@gmail.com (E.B.)

**Keywords:** autodyne, autodyne oscillator, autodyne signal, amplitude modulation, frequency modulation, short-range radar system

## Abstract

A mathematical model of an autodyne oscillator with a combined amplitude-frequency modulation, which is under the influence of its own radiation reflected from the target, is developed. The main relations are obtained for analyzing the autodyne response of a single-circuit oscillator depending on the delay time of the radiation reflected from the target with an arbitrary law of simultaneous amplitude and frequency modulation. The characteristics of the amplitude selection of signals, as well as their temporal and spectral characteristics, are calculated for harmonic amplitude-frequency modulation. The features of signal formation are established for various values of the amplitude and frequency modulation parameters and the feedback parameter of the “oscillator—target” autoparametric system. The results of experimental studies, performed on a Tigel-0.8M hybrid-integrated Ka-band oscillator module with a Gann mesa-planar diode, confirmed the findings of the theoretical analysis.

## 1. Introduction

Short-range radar sensors (SRRS) are used in control and safety systems in transport, security alarms, robotics, medicine, military affairs, and other fields [1,2,3,4,5,6,7]. The simplest design, minimum dimensions, lowest weight and lowest cost for SRRS are provided by autodyne transceivers. In short-range autodyne sensors (SRAS), the functions of the transmitter and receiver are performed by a single device connected directly to the antenna [8]. This device is usually based on an oscillator, which is known as an autodyne (AD). It generates probing microwave oscillations that are emitted by the antenna in the direction of the target. The radio signal reflected from the target is received by the same antenna and enters the oscillator system of the oscillator, generating a complex nonlinear process commonly called the autodyne effect [9,10,11,12,13,14,15,16].

This effect is manifested in changes in almost all oscillation parameters of the oscillator. It is observed in all types of oscillators, both continuous radiation and with various types of modulation, and in all ranges from radio frequencies to optical frequencies [17,18,19,20]. Not only are high-frequency oscillation parameters (amplitude, frequency, phase, and output power) subject to change: so are low-frequency parameters, such as direct currents and voltages in the oscillator bias circuits. Registration of these changes in the form of autodyne signals and their processing provide the opportunity to obtain the necessary information about the kinematic and electrophysical parameters of the targets, their radar cross-section (RCS), and the characteristics of the propagation medium of microwave radiation and antenna systems [21,22,23,24,25,26,27].

Due to the dependence of the oscillator parameters on the intrinsic microwave radiation reflected from the target, the AD belongs to the class of autoparametric systems with delayed feedback [28]. When analyzing these devices, separate consideration of the functions of generating and transmitting probe radiation, as well as receiving and converting radiation reflected from the target, is impossible.

This combination creates the problem of choosing the optimal oscillation mode in which the best characteristics of AD as a radar are provided, since the optimal modes can vary significantly for each of the indicated oscillator functions. The complexity of solving this problem, which lies in using methods from the theory of nonlinear oscillations, is often an obstacle to the successful use of AD in solving many short-range radar problems.

The operating conditions of SRRS are such that their range is usually commensurate with the target’s geometric dimensions. A reflected radio signal under these conditions is received from a plurality of brilliant points in the target. SRRS is also characterized by the presence of foreign objects and the target itself in the near zone of the antenna. Variations in the range to the target and its RCS are much wider than conventional ‘long-range’ radars. In many applications, the duration of the target in the SRRS area and the processing time of the received signals for issuing commands are usually very limited. In this case, it is necessary to ensure target selection against a background of not only passive, but also active interference. Under the indicated operating conditions in SRRS, errors in fixing the position of the target arise, which are also commensurate with the distance. Therefore, in many cases, when creating SRRS, the specific task is to form so-called ‘dead zones’ and ‘target selection zones’ by choosing the type and parameters of radiation modulation, as well as the corresponding processing of the received signal.

In addition to the general operating conditions of SRRS mentioned above, when developing SRAS, it is necessary to take into account a number of specific properties and features of AD operation. Among them is the inertia of the autodyne effect, which limits the speed of the system. If the oscillator parameters are improperly selected, the inertial properties of AD, characterized by the equivalent time constant of the autodyne response, can limit the operating frequency range of SRAS [17]. Another feature is the presence of the anharmonic distortion of signals increasing with the shortening of radiation wavelength and creating problems during processing [19].

It is possible to expand the functionality and improve the operability of SRAS in difficult radar conditions when using AD with amplitude (AM) or frequency (FM) modulation with continuous emission of radio waves [15,29,30]. However, in reality, when an oscillator is modulated by one type of modulation, another type of undesired modulation inevitably arises. For example, with AM of solid-state microwave oscillators along the power supply chain, FM is inevitably present, which imposes its conditions on the formation of the oscillator’s autodyne response. The influence of these two modulations at the same time on the formation of the autodyne response has not been fully considered in the literature. The task of researching new modes and searching for original technical solutions in order to increase noise immunity, improve other quality indicators and expand the scope of AD are certainly relevant and require resolution.

The purpose of this work is to fill the indicated gap by studying the formation of an autodyne signal with simultaneous amplitude and frequency modulation (AFM) of the oscillator oscillations. These features must be considered when creating advanced microwave and millimeter-wave SRAS.

## 2. General Equations for Describing the Autodyne Effect in Oscillators with AFM

The impact on the oscillator of the reflected radiation causes an autodyne effect, consisting of changes in the current values of the amplitude and frequency of the oscillations relative to their stationary values and the autonomous oscillator. The result of such an impact in the framework of the AD model in the form of a single-circuit oscillator on an active element with negative conductivity is described by a system of linearized differential equations with a delayed argument for small relative changes in the amplitude a and frequency χ of oscillations [30]
(1)$$(Q_{L}/\omega_{c})(da/dt)+\alpha a+\varepsilon \chi =\Gamma (t,\tau )\eta \cos\delta (t,\tau ),$$
(2)$$\beta a+Q_{L}\chi =-\Gamma (t,\tau )\eta \sin\delta (t,\tau ),$$
where $Q_{L}$ is the loaded Q factor and the frequency of the resonator; α, ε, β are dimensionless parameters that determine the reduced steepness of the increment, the nonisodromy and nonisochronism of the oscillator, respectively [29,30]; $\Gamma (t,\tau )=\Gamma_{0}[A(t,\tau )/A(t)]$, δ(t,τ)=Ψ(t)− Ψ(t,τ) are the modulus and phase of the instantaneous reflection coefficient reduced to the oscillator load; A(t,τ), Ψ(t,τ) are the amplitude and phase of the oscillations of AD from the history of the system (t–τ); A(t), Ψ(t) are the amplitude and phase of the oscillations of AD at the current time; $\Gamma_{0}={\left(P_{\mathrm{rec}}/P_{\mathrm{rad}}\right)}^{1/2}$ is the attenuation coefficient of the radiation in amplitude during its propagation to the target and vice versa; $P_{\mathrm{rad}}$, $P_{\mathrm{rec}}$ are the radiated and received power of microwave radiation at the oscillator load; $\eta =Q_{L}/Q_{\mathrm{ex}}$, $Q_{\mathrm{ex}}$ are the efficiency and external quality factor of the oscillatory system; τ=2l/c is the delay time of the reflected radiation; l is distance to the target; and c is the speed of propagation of radio waves.

The mathematical model of AD in the form of systems (1) and (2), which is based on the amplitude-phase delay of the radiation reflected from the target, is also valid for the case of AD with AFM. In this model, the expression for the quasi-harmonic output of AFM oscillations emitted by the antenna in the direction of the target can be written in a general form as
(3)$$u_{\mathrm{rad}}(t)=A(t)\;\cos\Psi (t)=A_{0}[1+m_{\mathrm{AM}}f_{m}(t)]\;\cos\;\left[\omega_{0}t+\omega_{0}m_{\mathrm{FM}}\int_{0}^{t}f_{m}(t)dt\right],$$
where $m_{\mathrm{AM}}=\Delta A_{\mathrm{AM}}/A_{0}$, $m_{\mathrm{FM}}=\Delta \omega_{\mathrm{FM}}/\omega_{0}$ are the coefficients of AM and FM oscillations; $\Delta A_{\mathrm{AM}}$, $\Delta \omega_{\mathrm{FM}}$ are maximum deviations of the amplitude and frequency of the oscillations from their stationary values $A_{0}$ and $\omega_{0}$ due to modulation; and $f_{m}(t)$ is the modulating function.

The oscillations reflected from the target and received by the antenna can be written as
(4)$$\begin{array}{c} u_{\mathrm{rec}}(t,\tau )=\Gamma_{0}A(t,\tau )\;\cos\Psi (t,\tau )\\ =\Gamma_{0}A_{0}[1+m_{\mathrm{AM}}f_{m}(t,\tau )]\;\cos\;\left[\omega_{0}(t-\tau )+\omega_{0}m_{\mathrm{FM}}\int_{t-\tau }^{t}f_{m}(t,\tau )dt+\varphi_{0}\right], \end{array}$$
where $f_{m}(t,\tau )$ is the modulating function of the reflected oscillations and $\varphi_{0}$ is a constant phase shift due to the reflective properties of the target.

The solution of system (1), (2), taking into account (3), (4) for low-signal autodyne changes in the amplitude A(t) and frequency ω(t) of oscillations of the oscillator when $\Gamma_{0}<<1$ in a quasistatic approximation, has the form
(5)$$A(t)=A_{0}\{1+m_{\mathrm{AM}}f_{m}(t)+\Gamma (t,\tau )K_{a}\cos[\delta (t,\tau )-\psi ]\},$$
(6)$$\omega (t)=\omega_{0}\{1+m_{\mathrm{FM}}f_{m}(t)-\Gamma (t,\tau )L_{a}\sin[\delta (t,\tau )+\theta ]\},$$
where $K_{a}$, $L_{a}$ are the coefficients of autodyne amplification and the deviation of the generation frequency [29,30]; ψ=arctg(ω), θ=arctg(γ) are the phase shift angles; and $\rho =\varepsilon /Q_{L}$, γ=β/α are the non-isodromic and non-isochronous coefficients of the oscillator, respectively.

The second term on the right side of Equation (5) repeats the modulation law $m_{\mathrm{AM}}f_{m}(t)$ and determines the level of the concomitant, so-called ‘parasitic’ AM (PAM) when registering the autodyne signal by changing the amplitude of the oscillations. At a low PAM level, the influence of this term on the formation of a useful signal can be neglected. However, in some cases, for example, when the oscillator is modulated by changing the supply voltage, PAM can have a significant and undesirable effect on the formation of the autodyne response.

Let us analyze the features of signal formation for the continuous AFM radiation regime described in (5) and (6) by the third members of the right-hand side. To identify the functions of the delayed action of $f_{m}(t,\tau )$, A(t,τ), and Ψ(t,τ) in these expressions, we perform their expansion into a Taylor series in the small parameter τ compared to the current time [31,32]
(7)$$f_{m}(t,\tau )=f_{m}(t)+\sum_{n=1}^{N}{\left(-1\right)}^{n}\frac{\tau^{n}}{n!}\frac{d^{n}f_{m}(t)}{dt^{n}},$$
(8)$$\Gamma (t,\tau )=\Gamma_{0}\frac{A(t,\tau )}{A(t)}=\Gamma_{0}\left[1-\frac{m_{\mathrm{AM}}}{1+a(t)}M_{\mathrm{AM}}(t)+\frac{a_{m}}{1+a(t)}M_{\mathrm{AD}}(t)\right],$$
(9)$$\delta (t,\tau )=\Psi (t)-\Psi (t,\tau )=\omega_{0}\tau +p_{\mathrm{PM}}F_{\mathrm{FM}}(t)-C_{\mathrm{FB}}F_{\mathrm{AD}}(t)$$
where n is, hereinafter, the serial number of the decomposition members and $M_{\mathrm{AM}}(t)$, $M_{\mathrm{AD}}(t)$ are the factors due to the AM process and autodyne changes in the oscillation amplitude of the oscillator, respectively:

(10)$$M_{\mathrm{AM}}(t)=\sum_{n=1}^{N}{\left(-1\right)}^{n-1}\frac{\tau^{n}}{n!}\cdot \frac{d^{n}f_{m}(t)}{dt^{n}},$$(11)$$M_{\mathrm{AD}}(t)=\sum_{n=1}^{N}{\left(-1\right)}^{n+1}H_{n}(\Omega_{a}\tau )\;\sin\left[\delta (t,\tau )-\psi -\Phi_{n}(\Omega_{a}\tau )\right].$$*F*_FM_(*t*), $F_{\mathrm{AD}}(t)$ are the factors due to the FM process and autodyne changes in the oscillation frequency of the oscillator, respectively
(12)$$F_{\mathrm{FM}}(t)=\sum_{n=1}^{N}{\left(-1\right)}^{n-1}\frac{\tau^{n-1}}{n!}\cdot \frac{d^{n-1}f_{m}(t)}{dt^{n-1}},$$
(13)$$F_{\mathrm{AD}}(t)=\sum_{n=1}^{N}{\left(-1\right)}^{n-1}S_{n}(\Omega_{a}\tau )\;\sin\left[\delta (t,\tau )+\theta -\Phi_{n}(\Omega_{a}\tau )\right],$$$a(t)=m_{\mathrm{AM}}f_{m}(t)+a_{m}\cos[\delta (t,\tau )-\psi ]$; $\Omega_{a}=d[\delta (t,\tau )]/dt$ is the instantaneous frequency of the autodyne signal; $a_{m}=\Gamma_{0}K_{a}$ is the relative depth of the autodyne changes in the amplitude of the oscillations; $p_{\mathrm{PM}}=m_{\mathrm{FM}}\omega_{0}\tau$ is the phase modulation index due to FM; $C_{\mathrm{FB}}={\Delta \omega }_{a}\tau$ is the external feedback parameter (FB) of AD; $\Delta \omega_{a}=\Gamma_{0}L_{a}\omega_{0}$ is the autodyne deviation of the oscillation frequency; and $H_{n}(\Omega_{a}\tau )$, $S_{n}(\Omega_{a}\tau )$ are the coefficients and $\Phi_{n}(\Omega_{a}\tau )$ are their phases
(14)$$H_{n}(\Omega_{a}\tau )=\frac{{\left(\Omega_{a}\tau \right)}^{\;2n-1}}{(2n-1)!}\sqrt{1+{\left(\frac{\Omega_{a}\tau }{2n}\right)}^{2}},S_{n}(\Omega_{a}\tau )=\frac{{\left(\Omega_{a}\tau \right)}^{\;2(n-1)}}{(2n-1)!}\sqrt{1+{\left(\frac{\Omega_{a}\tau }{2n}\right)}^{2}},$$
(15)$$\Phi_{n}(\Omega_{a}\tau )=\mathrm{arctg}\;{(\Omega }_{a}\tau /2n).$$

From expressions (9) and (13) it can be seen that in relation to δ(t,τ) the transcendental, Equation (8) is implicit. To identify (9), provided that it is smooth, when $C_{\mathrm{FB}}<1$, we find the δ(t,τ) by successive approximations. This solution in the form of an AD phase characteristic δ(t,τ) has the form
(16)$$\begin{array}{c} \delta (t,\tau )=\delta {\left(t,\tau \right)}_{\;(0)}-C_{\mathrm{FB}}\sum_{n=1}^{N}{\left(-1\right)}^{n-1}S_{n}(\Omega_{a}\tau )\;\sin[\delta {(t,\tau )}_{\;(1)}+\theta -\Phi_{n}(\Omega_{a}\tau )\\ -C_{\mathrm{FB}}\sum_{n=1}^{N}{\left(-1\right)}^{n-1}S_{n}(\Omega_{a}\tau )\;\sin[\delta {(t,\tau )}_{\;(2)}+\theta -\Phi_{n}(\Omega_{a}\tau )\\ -C_{\mathrm{FB}}\sum_{n=1}^{N}{\left(-1\right)}^{n-1}S_{n}(\Omega_{a}\tau )\;\sin[\delta {(t,\tau )}_{\;(k)}+\theta -\Phi_{n}(\Omega_{a}\tau )]]\ldots ], \end{array}$$
where $\delta {\left(t,\tau \right)}_{\;(0,1,\ldots k)}=\omega_{0}\tau +p_{\mathrm{PM}}F_{\mathrm{FM}}(t)$; the indices in parentheses around the terms indicate δ(t,τ) the approximation order.

In the analysis of ordinary ADs without modulation, it is necessary to put $m_{\mathrm{AM}}=p_{\mathrm{PM}}=0$ in (8) and (9). If, in this case, a strong inequality $\Omega_{a}\tau <<1$ holds, which is typical of most SRAS applications, then from (8), (9) we obtain $\Gamma (t,\tau )=\Gamma_{0}$ and δ(t,τ)=ωτ [19]. For the case of analysis of AD with FM in these expressions, $m_{\mathrm{AM}}$ should be equal to zero [30], and for the analysis of AD with AM it is necessary to put $m_{\mathrm{FM}}=0$ [29]. Thus, the developed mathematical model of AD is more general than the models in previous works.

Below, based on the obtained system of Equations (5) and (6) and expansions (7)–(15), we performed a numerical analysis of the behavior of AD with AFM for the harmonic law of the modulating function using the MathCAD program. When performing these calculations, approximations are accepted that take into account the real operating conditions of SRAS in the microwave and millimeter ranges.

## 3. Calculation and Analysis of Autodyne Characteristics at Sinusoidal AFM

AD signals are usually recorded in the oscillator power circuit (auto-detection signal) or by changing the amplitude of oscillations using a detector diode [33,34]. Changes in the amplitude and phase δ(t,τ) due to the modulation process and autodyne changes in the generation parameters, as well as the movement of the target, contribute to the formation of these signals. Below, we consider the case of sinusoidal modulation of the amplitude and frequency. The main advantages of this type of modulation are the scope of implementation and the relatively low level of higher harmonics of the modulation signal. In this regard, the selection of weak signals against the background of the signals’ main components is quite simple. Therefore, the harmonic law of the modulating function, as noted above, is widely used in autodyne systems to solve many short-range radar problems.

For normal operation of SRAS with AFM, the modulation frequency $\Omega_{m}$ should be significantly different from the frequency $\Omega_{a}$ of the autodyne signal. Under such conditions, the signal spectrum and spectral components due to the AFM process do not overlap, which simplifies the frequency selection of useful components in the autodyne signal’s spectrum [15]. It is of practical interest to consider the case of strong inequality $\Omega_{m}>>\Omega_{a}$, when the components of the useful signal are grouped both in the region of low “zero” frequencies and in the vicinity of harmonics of the modulation frequency $\Omega_{m}$ [29].

At the same time, we will discover the laws governing the formation of the useful signal’s amplitude depending on the distance to the target. This dependence is usually called the characteristic of the amplitude selection (CAS) of the target in range. The main equations for the analysis of such a characteristic are expressions (5) and (8), taking into account (9) and expansions (10)–(16). In these expressions, the real ratio of *m*_AM_ and $a_{m}$ is such that usually $m_{\mathrm{AM}}>>a_{m}$ and the influence of autodyne changes in the amplitude of oscillations on the level of reflected radiation can be neglected. Therefore, expression (5), with allowance for (8) in the calculation of CAS, can be significantly simplified by excluding the third term in (8) from further analysis. This expression for the harmonic law $f_{m}(t)=\sin\;{(\Omega }_{m}t)$ of the modulating function after the normalization of the amplitude relative to the product has the form
(17)$$a(t)=\left[1-\frac{m_{\mathrm{AM}}M_{\mathrm{AM}}(t)}{1+m_{\mathrm{AM}}\sin\;{(\Omega }_{m}t)}\right]\cos[\delta (t,\tau )-\psi ],$$
where
(18)$$M_{\mathrm{AM}}(t)=\sum_{n=1}^{N}{\left(-1\right)}^{n-1}M_{n}(r_{\mathrm{nd}})\cos\left[\left(\Omega_{m}t)-\Psi_{n}(r_{\mathrm{nd}}\right)\right],$$

$M_{n}(r_{\mathrm{nd}})$ are the coefficients and $\Theta_{n}(r_{\mathrm{nd}})$ are the phase angles of the expansion terms (17):(19)$$M_{n}(r_{\mathrm{nd}})=\frac{{\left(2\pi r_{\mathrm{nd}}\right)}^{\;2n-1}}{(2n-1)!}\sqrt{1+{\left(\frac{2\pi r_{\mathrm{nd}}}{2n}\right)}^{2}},\Psi_{n}(r_{\mathrm{nd}})=\mathrm{arctg}\;\frac{2\pi r_{\mathrm{nd}}}{2n},$$$r_{\mathrm{nd}}=\Omega_{m}\tau /2\pi =l/(\Lambda_{m}/2)$ is the normalized distance to the target and $\Lambda_{m}=2\pi c/\Omega_{m}$ is the modulating wavelength.

Here we rewrite the expression for the phase characteristic δ(t,τ) (17) in (16), taking into account the normalization of the distance to the target
(20)$$\begin{array}{c} \delta (t,\tau )={\left(2\pi \tau_{\mathrm{nt}}\right)}_{\left(0\right)}+2\pi k_{\mathrm{FM}}r_{\mathrm{nd}}F_{\mathrm{FM}}(t)-(C_{\mathrm{FB}}/2\pi )\sum_{n=1}^{N}{\left(-1\right)}^{n-1}S_{n}(r_{\mathrm{nd}})\;\sin[{\left(2\pi \tau_{\mathrm{nt}}\right)}_{\left(1\right)}+2\pi k_{\mathrm{FM}}r_{\mathrm{nd}}F_{\mathrm{FM}}(t)\\ +\theta -\Phi_{n}(r_{\mathrm{nd}})-(C_{\mathrm{FB}}/2\pi )\sum_{n=1}^{N}{\left(-1\right)}^{n-1}S_{n}(r_{\mathrm{nd}})\;\sin[{\left(2\pi \tau_{\mathrm{nt}}\right)}_{\left(2\right)}+2\pi k_{\mathrm{FM}}r_{\mathrm{nd}}F_{\mathrm{FM}}(t)+\theta -\Phi_{n}(r_{\mathrm{nd}})\\ -(C_{\mathrm{FB}}/2\pi )\sum_{n=1}^{N}{\left(-1\right)}^{n-1}S_{n}(r_{\mathrm{nd}})\;\sin[{\left(2\pi \tau_{\mathrm{nt}}\right)}_{\left(k\right)}+2\pi k_{\mathrm{FM}}r_{\mathrm{nd}}F_{\mathrm{FM}}(t)+\theta -\Phi_{n}(r_{\mathrm{nd}})]]\ldots ], \end{array}$$
where $\tau_{\mathrm{nt}}=\omega_{0}\tau /2\pi$ is the normalized time; $S_{n}(r_{\mathrm{nd}})$ are the coefficients; and $\Phi_{n}(r_{\mathrm{nd}})$ are the phase angles of the expansion terms (20)
(21)$$S_{n}(r_{\mathrm{nd}})=\frac{{\left(2\pi r_{\mathrm{nd}}/z\right)}^{\;2(n-1)}}{(2n-1)!}\sqrt{1+{\left(\frac{2\pi r_{\mathrm{nd}}}{2nz}\right)}^{2}},\;\Phi_{n}(r_{\mathrm{nd}})=\mathrm{arctg}\;\frac{2\pi r_{\mathrm{nd}}}{2nz},$$$z=\Omega_{m}/\Omega_{a}$ is the ratio of modulation frequencies and autodyne signal;
(22)$$F_{\mathrm{FM}}(t)=\sin(\Omega_{m}t)-\sum_{n=1}^{N}{\left(-1\right)}^{n-1}F_{n}(r_{\mathrm{nd}})\cos\left[\left(\Omega_{m}t)-\Theta_{n}(r_{\mathrm{nd}}\right)\right],$$$F_{n}(r_{\mathrm{nd}})$ are the coefficients and $\Theta_{n}(r_{\mathrm{nd}})$ are the phase angles of the expansion terms (22)
(23)$$F_{n}(r_{\mathrm{nd}})=\frac{{\left(2\pi r_{\mathrm{nd}}\right)}^{\;2n-1}}{2n!}\sqrt{1+{\left(\frac{2\pi r_{\mathrm{nd}}}{2n+1}\right)}^{2}},\;\Theta_{n}(r_{\mathrm{nd}})=\mathrm{arctg}\frac{2\pi r_{\mathrm{nd}}}{2n+1}.$$

From analysis (17), taking into account (18)–(23), it follows that CAS is a periodic function of the normalized distance $r_{\mathrm{nd}}$ to the target, determined by the period of the modulating function $f_{m}(t)$. Therefore, we further restrict ourselves to analyzing only its main component, enclosed in the interval $0<r_{\mathrm{nd}}<1$. In addition, to facilitate the analysis, we assume that z is a natural series of numbers; moreover, z>>1.

If the operating conditions of SRAS are such that it is necessary to consider CAS only near SRAS (the initial portion of CAS), where the inequality $r_{\mathrm{nd}}<<1$ holds, then expression (17) is simplified
(24)$$a(t)=\left[1-\frac{2\pi r_{\mathrm{nd}}m_{\mathrm{AM}}\cos\;{(\Omega }_{m}t)}{1+m_{\mathrm{AM}}\sin\;{(\Omega }_{m}t)}\right]\cos[2\pi k_{\mathrm{FM}}r_{\mathrm{nd}}\sin\;{(\Omega }_{m}t)].$$

However, with increasing distance to the target, where the use of expression (24) becomes unacceptable, the number of terms required for calculating the terms of the series in (18)–(23) grows rapidly. This greatly complicates the analysis. For example, if condition $r_{\mathrm{nd}}\leq 1$ is fulfilled, which means that the delay time τ of the reflected radiation should be no more than a sixth of the half-period of the modulating function, the required number of first terms of expansions (18)–(23) should be at least five. For a correct description of CAS over its entire interval $0<r_{\mathrm{nd}}<1$, as the calculations show, it is necessary to significantly increase the number N of members of this series: N≥25. Under such conditions, in the general case of arbitrary value $r_{\mathrm{nd}}$, clarification of the features of the formation of autodyne signals by analytical methods seems unacceptable; therefore, we will use the numerical method to achieve this goal.

Figure 1, Figure 2, Figure 3 and Figure 4 show the results of CAS calculations as a normalized function of two variables: the amplitude modulation coefficient $m_{\mathrm{AM}}$ (axis of ‘modulation’) and normalized to half the length of the modulating wave of the distance to the target $r_{\mathrm{nd}}$ (axis of ‘distance’). The vertical levels on these CAS are the relative levels $a_{\mathrm{rl}}(k)$ of the components of the autodyne response at the k-th harmonic of the modulation frequency $\Omega_{m}$. At the ‘zero’ harmonic, this axis is designated as $a_{\mathrm{rl}}\left(0\right)$ (see Figure 1). At the first harmonic of the modulation frequency, this is indicated as $a_{\mathrm{rl}}\left(1\right)$ (see Figure 2), at the second—$a_{\mathrm{rl}}\left(2\right)$ (see Figure 3), and the third—$a_{\mathrm{rl}}\left(3\right)$ (see Figure 4). The calculations were performed for the normalized useful signal $a_{\mathrm{rl}}=a/\Gamma_{0}K_{a\;}$ in the expansion of function (17), taking into account (18)–(23) in the harmonic Fourier series over the period of the modulating function $f_{m}(t)$. For the same CAS, Figure 5 and Figure 6 show families of graphs of their cross sections for $m_{\mathrm{AM}}=0.8$ (see Figure 5) and $r_{\mathrm{nd}}=0.5$ (see Figure 6).

For the case of a uniform change in the normalized time $\tau_{\mathrm{nt}}$, Figure 7 and Figure 8, respectively, show the results of calculating the time $a_{\mathrm{rl}}(\tau_{\mathrm{nt}})$ and spectral diagrams $a_{\mathrm{rl}}(F_{\mathrm{nf}})$ of an autodyne signal received for an oscillator with AM ($m_{\mathrm{AM}}=0.5$, $k_{\mathrm{FM}}=0$), $C_{\mathrm{FB}}=0.5$ and AFM ($m_{\mathrm{AM}}=0.5$, $k_{\mathrm{FM}}=1$). Here, the $F_{\mathrm{nf}}=\Omega /\Omega_{m}$ is the normalized frequency. The diagrams were calculated at a ratio z=10 for frequencies and displacement angles θ=1 and ψ=0.1 for different values of the normalized distance $r_{\mathrm{nd}}$ (see Figure 7 and Figure 8a–e).

From the characteristics and graphs of Figure 1, Figure 2, Figure 3 and Figure 4 under letters a and b, as well as curves 1 and 2 in Figure 5 shows that the maximum autodyne response in the absence and relatively ‘low’ level of FM corresponds to the middle of the amplitude selection zone when $r_{\mathrm{nd}}=0.5$. This behavior of the autodyne response in the middle of CAS when the AM process dominates over FM is explained by the phenomenon of signal regeneration in the ‘oscillator—target—oscillator’ feedback loop, when the process of AM oscillations and the influence of the reflected radiation amplitude are in phase.

Such phenomena are observed, for example, in short-range “recirculation” meters [35,36]. In these meters, the voltage of the reflected signal modulation, allocated by the SRAS receiver, after amplification is again supplied in the required polarity to the transmitter modulator. By choosing the parameters of the elements of the closed loop (amplification and phase shift), which includes the target and the propagation medium, it is possible to achieve the generation mode at a certain frequency, which depends on the delay time of the radiation reflected from the target. It can be seen from the results of CAS calculations that for harmonic AM and a low FM level, it is most beneficial to use large values of the coefficient of AM, the value of which approaches unity. In this case, it becomes possible to isolate signals at the higher harmonics of the modulation frequency.

In the case of a ‘high’ FM level, the region of the main maximum shifts to the side where $r_{\mathrm{nd}}>0.5$, and the appearance of a multi-humped CAS is noted (see Figure 1, Figure 2, Figure 3 and Figure 4 under letter c and curve 2 in Figure 5). In this case, the response amplitude $\Omega_{m}$ at all harmonics of the modulation frequency, including ‘zero’, in all cases of the presence or absence of concomitant FM increases significantly with the coefficient $m_{\mathrm{AM}}$ approaching unity (see Figure 6a–d). It is also interesting to note that in the absence of AM, when $m_{\mathrm{AM}}=0$ but the presence of FM is at zero and even harmonics $\Omega_{m}$, the formation of a multi-humped CAS is observed.

The CAS of the autodyne response at harmonics of the modulation frequency differ significantly from CAS formed in the region of ‘zero’ (Doppler) frequencies (see Figure 1, Figure 2, Figure 3 and Figure 4). This difference consists in the presence of ‘dead’ zones in CAS at harmonics, where signals reflected from targets located at certain distances from SRAS are suppressed. One of these critical areas is in close proximity to SRAS, where $r_{\mathrm{nd}}=0$. In this zone, the appearance of relatively small reflecting objects, such as insects, raindrops and others, causes the appearance of powerful reflected signals, which in some applications disrupt the normal operation of SRAS. In this case, the presence of a dead zone near an SRAS antenna with AFM when isolating signals at the harmonics of the modulation frequency is a very important property of these systems, providing increased noise immunity.

From the results of the calculations, it is clear that in SRAS with AM, according to the sinusoidal law, it is most expedient to use large values of the AM coefficient. At the same time, the ability is obtained to select signals at higher harmonics of the modulation frequency at which the CAS form is more preferable for a number of applications: the PAM level is also much lower. However, with harmonic AM, as noted above, the signal level at higher harmonics decreases sharply with the increasing number. Therefore, the use of harmonics above the third order becomes irrational. In this case, the presence of concomitant FM radiation of AD contributes to a change in the ratio between harmonic levels in favor of increasing their order.

Timing diagrams, as seen from Figure 7, contain slow (Doppler) and fast components. The slow component in the ‘pure’ form shown in Figure 7a,e is similar to the amplitude characteristic of AD without modulation [37]. From these graphs, it can be noted that at the beginning and at the end of each CAS, the autodyne response at the AD output from AM is presented only as a Doppler signal. There is no response at the harmonics of the modulation frequency $\Omega_{m}$ in these cases. When the target is shifted to the middle CAS region, additional components in the form of ‘peaks’ are superimposed on the Doppler signal component due to the interaction of the reflected AM oscillations with the emitted ones, which are also ‘modulated’ in amplitude with frequency $\Omega_{m}$ (see Figure 7b–d).

These additional components cause a corresponding increase in the signal level at both the zero and harmonics of the modulation frequency $\Omega_{m}$. As the calculations showed, the amplitude of the peaks increases significantly with the coefficient $m_{\mathrm{AM}}$ approaching unity, as well as when passing to the region of the middle part of CAS, where $r_{\mathrm{nd}}=0.5$. From the spectrograms in Figure 7, it can be seen that the spectral components obtained in AD with AM at frequencies that are multiples of the modulation frequency $\Omega_{m}$ decompose into two components of half amplitude when the target moves. These components are offset from the frequency multiple $\Omega_{m}$ by the magnitude of the frequency $\Omega_{a}$ of the autodyne signal, equal to the Doppler frequency.

As can be seen from Figure 8a,e, at the beginning and end of each CAS, the autodyne response at the output of the autodyne with AFM is presented only as a Doppler signal, and there is no response at the harmonics of the modulation frequency $\Omega_{m}$ in these cases. In the middle part of CAS, where $0<r_{\mathrm{nd}}<1$, additional components associated with the FM process are superimposed on the signal components due to the AM process, as well as to signal transfer to the harmonics of the modulation frequency $\Omega_{m}$ (see Figure 7b–d). The phase differences of these components cause a noticeable asymmetry of the signal spectrum at the harmonics of the modulation frequency.

The presence of anharmonic distortions in the time diagrams and higher harmonics of the frequency of the autodyne signal in its spectrum (see Figure 7 and Figure 8 under letters a and e) is explained by the well-known nonlinearity of the phase incursion of the reflected radiation caused by autodyne changes in the generation frequency of AD [17,19]. The degree of these signal distortions is determined primarily by the value of the feedback parameter $C_{\mathrm{FB}}$. So, with $C_{\mathrm{FB}}<<1$ the autodyne changes are almost harmonic; with its increase, they significantly differ from sinusoidal ones, acquiring the ‘wave slope’ of the autodyne signal. The ‘direction’ of this slope depends on the sign and magnitude of the parameters of nonisochronism (through an angle θ) and nonisodromy (through an angle ψ), the oscillator and the relative direction of motion of the target. These distortions are the cause of the appearance of higher harmonics of the Doppler signal, both in the region of low (Doppler) frequencies and in the vicinity of harmonics of the modulation frequency $\Omega_{m}$.

## 4. Results of Experimental Studies

Experimental studies, the purpose of which is to verify the conclusions of the theoretical analysis, were performed using the Tigel-08M Ka-band oscillator module (see Figure 9a), manufactured using a hybrid-integrated technology based on a two-mesa Gunn diode [37]. On the substrate of this module, chips of a planar Gunn diode and a detector diode with a Schottky barrier, isolated on the bias circuits (shown in Figure 9b by the numbers ‘1’ and ‘2’, respectively), were installed in the ‘slot resonator’ in parallel (via the microwave). One of the mesa of the Gann diode was made of a large cross section. Its current density was insufficient to excite the domains of a strong field; therefore, it was passive. The second mesa, a small cross section, was active and created the conditions for the excitation of oscillations in the oscillator. The detector diode is designed for measuring the autodyne signal by changing the amplitude of the oscillations.

In the design of the Tigel-08M module, the substrate was placed between the two plates forming its body (see Figure 9a). A circular hole was provided in the center of the front plate for outputting microwave radiation, and the back plate was bound with a screw for adjusting the frequency. In order to stabilize the generation frequency, an additional high-Q resonator was docked to the rear wall of the module (see Figure 9c). The coupling between the working and stabilizing resonators was a through slot in the rear wall in the form of a segment of the waveguide channel. A stabilizing resonator made of superinvar and operating on a wave $H_{011}$ had its own Q factor of the order of 5 × 10^3^. The resistive coupling was provided by introducing a wedge-shaped absorber insert into the coupling window with the resonator, the position of which can be adjusted during tuning (see Figure 9d).

Previously, the module was tuned to the maximum autodyne sensitivity mode without frequency stabilization using a Doppler signal simulator. For this, the magnitude of the coupling was changed by varying the position of the screw provided for in the module design: the bias voltage $U_{0}$ on the Gunn diode was selected. The optimal value of this voltage for the module under study was chosen within $U_{0}=3.6...4$ V; the current consumption was 0.2 A. The generation frequency of the module was 37.5 GHz, while its output power was 15 mW.

When tuning to the maximum autodyne sensitivity mode of the stabilized module, the connection between the working and stabilizing resonators was also corrected by changing the position of the absorbing insert and the size of the coupling hole. This connection was established so that the output power of the oscillator was reduced by no more than 5–10% of the nominal value. The coincidence of the generation frequency with the natural frequency of the stabilizing resonator was monitored at the time of ‘frequency capture’, when the autodyne deviation of the generation frequency was minimal.

The functional diagram of the experimental setup is shown in Figure 10. It provides registration of the autodyne response when changing the distance to the reflector-simulator, its speed, level of the reflected signal, and other factors.

The studied autodyne oscillator AO (see Figure 10) was connected to the Doppler simulator DS [38] by the waveguide path (WP), which also contained a directional coupler (DC) and a variable attenuator (Att). The waveguide path of the required length is designed to simulate the delay of the reflected radiation, and the attenuator Att—to provide attenuation of the radiation propagating in space to the target and back. The lateral arm of the directional DC coupler was connected to the input of the frequency converter (FC), the output of which was connected to the input of the first spectrum analyzer (SA-1). The output signal of the built-in AO detector was fed to the input of the second spectrum analyzer (SA-2). Both USB-SA44B-type spectrum analyzers were connected to a PC using USB cables. Power from a stabilized power source (PS) to the AO oscillator under study was supplied through a modulator (Mod) made on a bipolar transistor. A sinusoidal voltage of 10 MHz came from the function generator (FG) to the control input of the modulator. In this case, the required AM coefficient of the AO was controlled by the SA-1.

The following conditions were realized in the experiments: the length of the WP with the cross-section 7.2×3.4 mm^2^ between the AO and the DS (see Figure 10) was $l_{w}=9.5$ m, i.e., the delay time of the reflected radiation in the waveguide, calculated by the formula $\;\tau =2l_{w}/c\sqrt{1-{\left(\lambda /\lambda_{\mathrm{cr}}\right)}^{2}}$, where $\lambda_{\mathrm{cr}}$—is the critical wavelength of the waveguide, was $\tau =75\times 10^{-9}$ s. This distance approximately corresponds to a selected modulation frequency of 10 MHz in the middle of CAS. A variable Att in the path set the value of the reflection coefficient $\Gamma_{0}$ (at which the autodyne frequency deviation was $\Delta \omega_{a}=2\omega \times 1.5\times 10^{6}$) and the calculated value of the feedback parameter $C_{\mathrm{FB}}=0.5$. The frequency deviation was estimated using the SA-1 with a working DS from the width of the microwave radiation spectrum of the AO.

In Figure 11a shows the microwave oscillation spectrum of the AO obtained from a frequency-stabilized module with an AM depth of about 50%. The value of AM was estimated by the ratio of the main and side components of the spectrum obtained using the SA-1 (see Figure 10). The spectrogram of microwave oscillations of the same unstabilized module at the previous depth of AM is shown in Figure 11b. The spectrum’s asymmetry in the latter case indicates the presence of concomitant FM.

Under the experimental conditions indicated above, using the SA-2 (see Figure 10), we obtained the spectra of autodyne signals for the stabilized and ordinary oscillator modules which are presented in Figure 12a,b, respectively. The speed of the simulator’s reflector was kept constant, at which the frequency of the autodyne signal was 10 kHz. The spectra show the regions of zero and the first three harmonics of the modulation frequency. The presence of higher harmonics of the Doppler frequency in these spectral regions is due to anharmonic distortion of the signals. These distortions, as has already been established [19], are observed when the value of the feedback parameter is comparable with unity. If this parameter decreases, for example, by increasing attenuation of Att by 10–20 dB the higher harmonics of the Doppler frequency practically disappear, and the signal becomes sinusoidal.

The spectra of Figure 12 also shows that the level of the first harmonic of PAM is significantly higher than the level of the harmonics of a higher order. Significant differences in the spectral shape of the investigated oscillator modules are the asymmetry of the spectra in the vicinity of the harmonics of the modulation frequency in the case of an unstabilized oscillator module, whose level of accompanying FM radiation is much higher than that of the stabilized module (see Figure 12b).

From a comparison of the spectral diagrams calculated above (presented in Figure 7 and Figure 8c) and the experimental spectra shown in Figure 12, their qualitative correspondence is demonstrated. Thus, the results of experimental studies confirm the adequacy of the developed mathematical model with respect to the influence of concomitant FM on the formation of AD signals with AM.

## 5. Conclusions

A mathematical model of an autodyne oscillator with AFM radiation has been developed, which provides the ability to calculate the CAS, shape and spectrum of the autodyne signal for the general case of an arbitrary ratio of the delay time of the radiation reflected from the target and the period of the modulating function. As a result of the calculations for the case of the harmonic modulation law and experimental studies of AD on Gunn diodes, the influence of the accompanying frequency modulation on the formation of autodyne signals was established.

In the absence of or at a relatively ‘low’ FM level, the maximum of the autodyne response corresponds to the middle of the amplitude selection zone. In the case of a ‘high’ FM level, the region of the main maximum shifts to the side of large values of the normalized distance $r_{\mathrm{nd}}$ and the appearance of a multi-humped CAS is noted. In this case, the response amplitude at all harmonics of the modulation frequency $\Omega_{m}$, including the ‘zero’ harmonic, in all cases of the presence or absence of concomitant FM, increases significantly with the amplitude modulation coefficient $m_{\mathrm{AM}}$ approaching unity.

Autodyne oscillators with AFM, as well as ordinary unmodulated ADs, are characterized by anharmonic distortion of signals due to autodyne changes in the generation frequency under conditions when the values of the feedback parameter are comparable with unity. The presence of these distortions requires consideration when choosing the parameters of autodyne oscillators and in signal processing devices.

A distinctive feature of AD with AFM is that the autodyne response is also transferred to the harmonics of the modulation frequency, and the efficiency of this transfer increases with increasing modulation depth: it also depends on the concomitant deviation of the generation frequency and the distance to the target. Due to this, ADs with AFMs at harmonics of modulation frequencies have the property of amplitude selection of targets at certain distances, which provides for the increased noise immunity of SRAS. It has also been found that, compared with conventional, unstabilized AD, the stabilized autodyne oscillator provides a significant reduction in the accompanying FM and the degree of signal distortion. In addition, the research results showed the advantages of AD with frequency stabilization over conventional, unstabilized ADs and the feasibility of developing and manufacturing new types of monolithic and hybrid integrated autodyne millimeter-wave modules stabilized by an additional high-Q resonator.

## Figures and Tables

**Figure 1 sensors-20-07077-f001:**
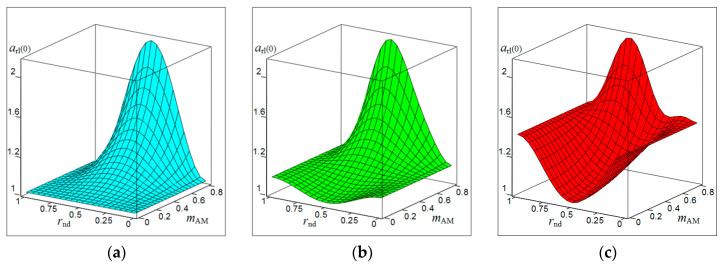
Characteristics of the amplitude selection (CAS) $a_{\mathrm{rl}}\left(0\right)$ of the autodyne signal on the variation of the oscillation amplitude with AFM on the zero harmonic of the modulation frequency calculated for different values of $k_{\mathrm{FM}}$: (**a**) $k_{\mathrm{FM}}=0$, (**b**) $k_{\mathrm{FM}}=0.4$, and (**c**) $k_{\mathrm{FM}}=0.8$.

**Figure 2 sensors-20-07077-f002:**
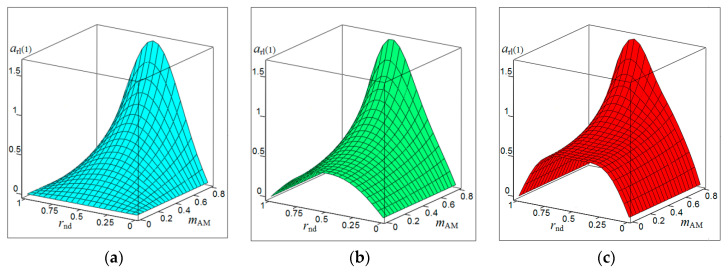
CAS $a_{\mathrm{rl}}\left(1\right)$ of the autodyne signal on the variation of the oscillation amplitude with AFM on the first harmonic of the modulation frequency calculated for different values of $k_{\mathrm{FM}}$: (**a**) $k_{\mathrm{FM}}=0$, (**b**) $k_{\mathrm{FM}}=0.5$, and (**c**) $k_{\mathrm{FM}}=1$.

**Figure 3 sensors-20-07077-f003:**
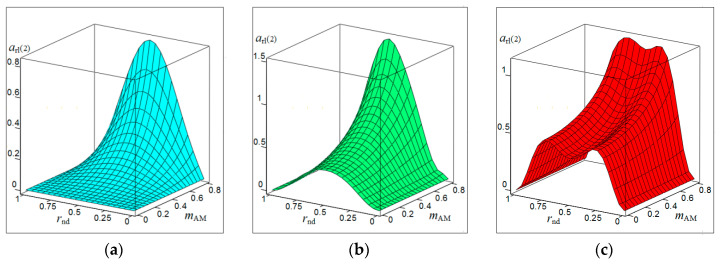
CAS $a_{\mathrm{rl}}\left(2\right)$ of the autodyne signal on the variation of the oscillation amplitude with AFM on the second harmonic of the modulation frequency calculated for different values of $k_{\mathrm{FM}}$: (**a**) $k_{\mathrm{FM}}=0$, (**b**) $k_{\mathrm{FM}}=1$, and (**c**) $k_{\mathrm{FM}}=2$.

**Figure 4 sensors-20-07077-f004:**
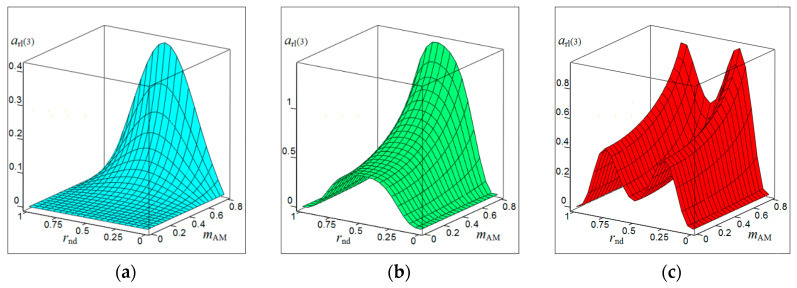
CAS $a_{\mathrm{rl}}\left(3\right)$ of the autodyne signal on the variation of the oscillation amplitude with AFM on the third harmonic of the modulation frequency calculated for different values of $k_{\mathrm{FM}}$: (**a**) $k_{\mathrm{FM}}=0$, (**b**) $k_{\mathrm{FM}}=2$, and (**c**) $k_{\mathrm{FM}}=3$.

**Figure 5 sensors-20-07077-f005:**
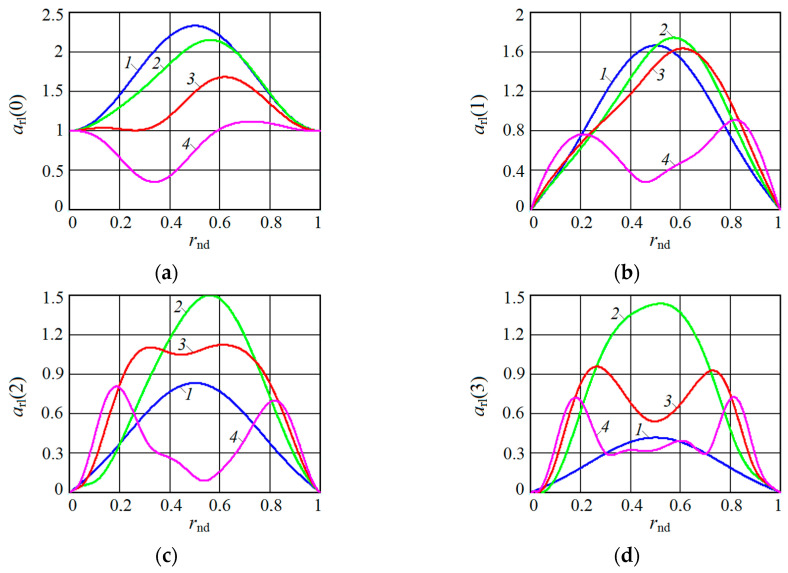
Plots of CAS section along the axis $r_{\mathrm{nd}}$ on the (**a**) zero $a_{\mathrm{rl}}\left(0\right)$, (**b**) first $a_{\mathrm{rl}}\left(1\right)$, (**c**) second $a_{\mathrm{rl}}\left(2\right)$ and (**d**) third $a_{\mathrm{rl}}\left(3\right)$ harmonics of the modulation frequency calculated at $m_{\mathrm{AM}}=0.8$ for different values of $k_{\mathrm{FM}}$. On plot **a**: $k_{\mathrm{FM}}=0$ (curve 1), $k_{\mathrm{FM}}=0.4$ (curve 2), $k_{\mathrm{FM}}=0.8$ (curve 3), $k_{\mathrm{FM}}=1.2$ (curve 4). On plot **b**: $k_{\mathrm{FM}}=0$ (curve 1), $k_{\mathrm{FM}}=0.5$ (curve 2), $k_{\mathrm{FM}}=1$ (curve 3), $k_{\mathrm{FM}}=2$ (curve 4). On plot **c**: $k_{\mathrm{FM}}=0$ (curve 1), $k_{\mathrm{FM}}=1$ (curve 2), $k_{\mathrm{FM}}=2$ (curve 3), $k_{\mathrm{FM}}=3$ (curve 4). On plot **d**: $k_{\mathrm{FM}}=0$ (curve 1), $k_{\mathrm{FM}}=2$ (curve 2), $k_{\mathrm{FM}}=3$ (curve 3), $k_{\mathrm{FM}}=4$ (curve 4).

**Figure 6 sensors-20-07077-f006:**
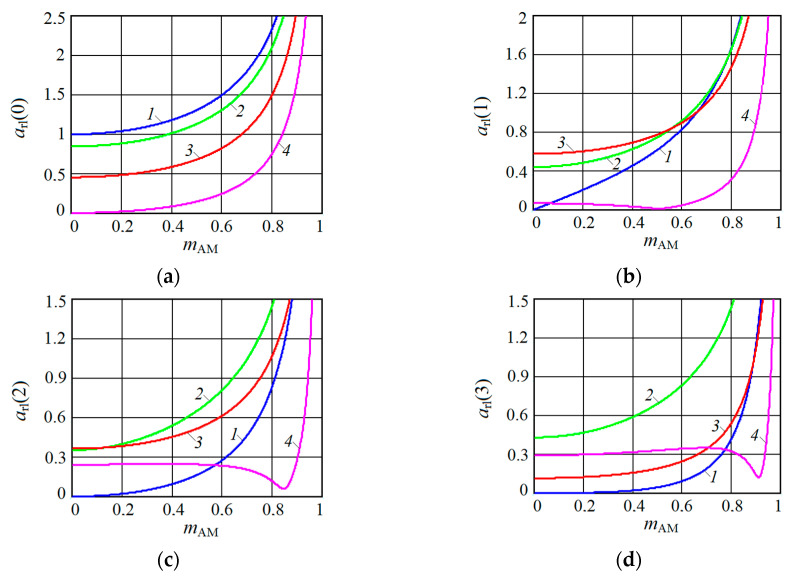
Plots of CAS sections along axis $m_{\mathrm{AM}}$ on (**a**) zero $a_{\mathrm{rl}}\left(0\right)$, (**b**) first $a_{\mathrm{rl}}\left(1\right)$, (**c**) second $a_{\mathrm{rl}}\left(2\right)$ and (**d**) third $a_{\mathrm{rl}}\left(3\right)$ harmonics of the modulation frequency calculated at $r_{\mathrm{nd}}=0.5$ for different values $k_{\mathrm{FM}}$. On plot **a**: $k_{\mathrm{FM}}=0$ (curve 1), $k_{\mathrm{FM}}=0.4$ (curve 2), $k_{\mathrm{FM}}=0.8$ (curve 3), $k_{\mathrm{FM}}=1.2$ (curve 4). On plot **b**: $k_{\mathrm{FM}}=0$ (curve 1), $k_{\mathrm{FM}}=0.5$ (curve 2), $k_{\mathrm{FM}}=1$ (curve 3), $k_{\mathrm{FM}}=2$ (curve 4). On plot **c**: $k_{\mathrm{FM}}=0$ (curve 1), $k_{\mathrm{FM}}=1$ (curve 2), $k_{\mathrm{FM}}=2$ (curve 3), $k_{\mathrm{FM}}=3$ (curve 4). On plot **d**: $k_{\mathrm{FM}}=0$ (curve 1), $k_{\mathrm{FM}}=2$ (curve 2), $k_{\mathrm{FM}}=3$ (curve 3), $k_{\mathrm{FM}}=4$ (curve 4).

**Figure 7 sensors-20-07077-f007:**
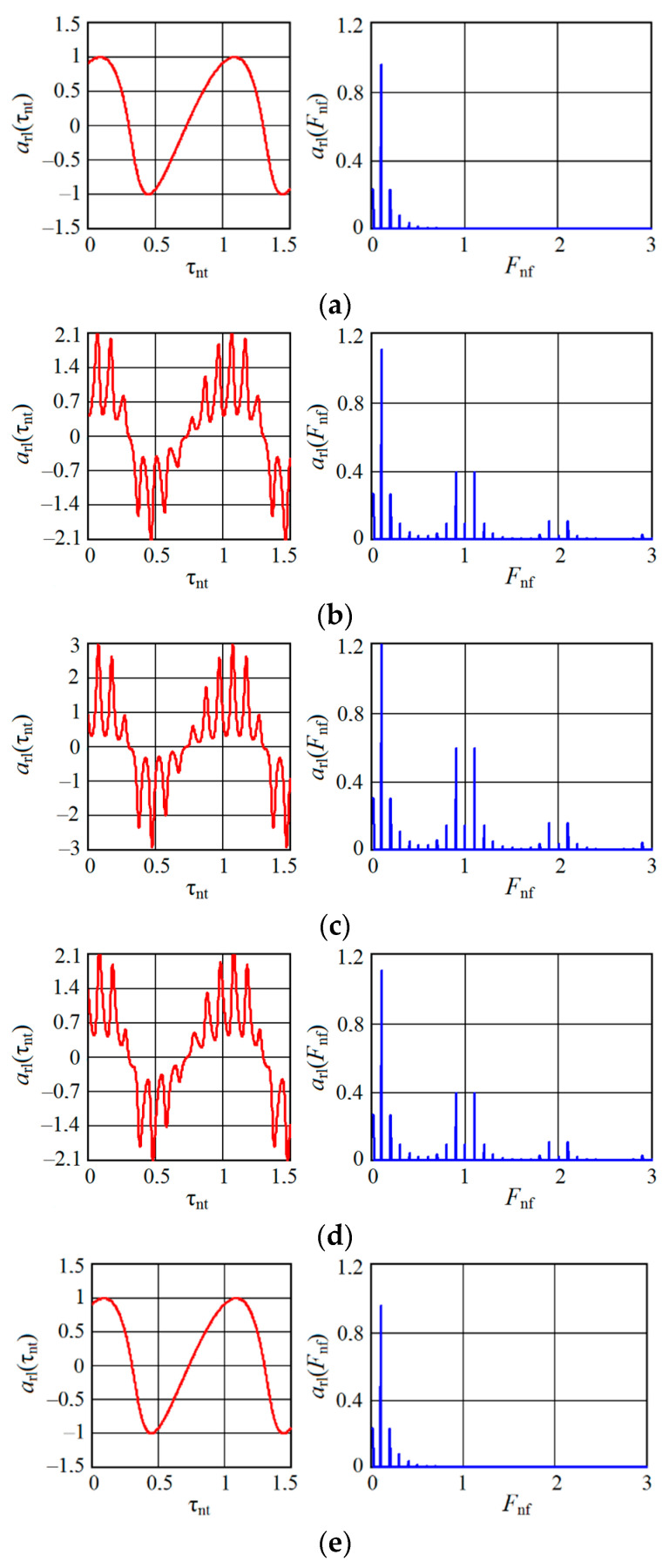
Time $a_{\mathrm{rl}}(\tau_{\mathrm{nt}})$ and spectral $a_{\mathrm{rl}}(F_{\mathrm{nf}})$ diagrams of the autodyne signal calculated for the oscillator with AM in absence of FM ($k_{\mathrm{FM}}=0$ ) and different values $r_{\mathrm{nd}}$: (**a**) $r_{\mathrm{nd}}=0$; (**b**) $r_{\mathrm{nd}}=0.25$; (**c**) $r_{\mathrm{nd}}=0.5$; (**d**) $r_{\mathrm{nd}}=0.75$; (**e**) $r_{\mathrm{nd}}=1$.

**Figure 8 sensors-20-07077-f008:**
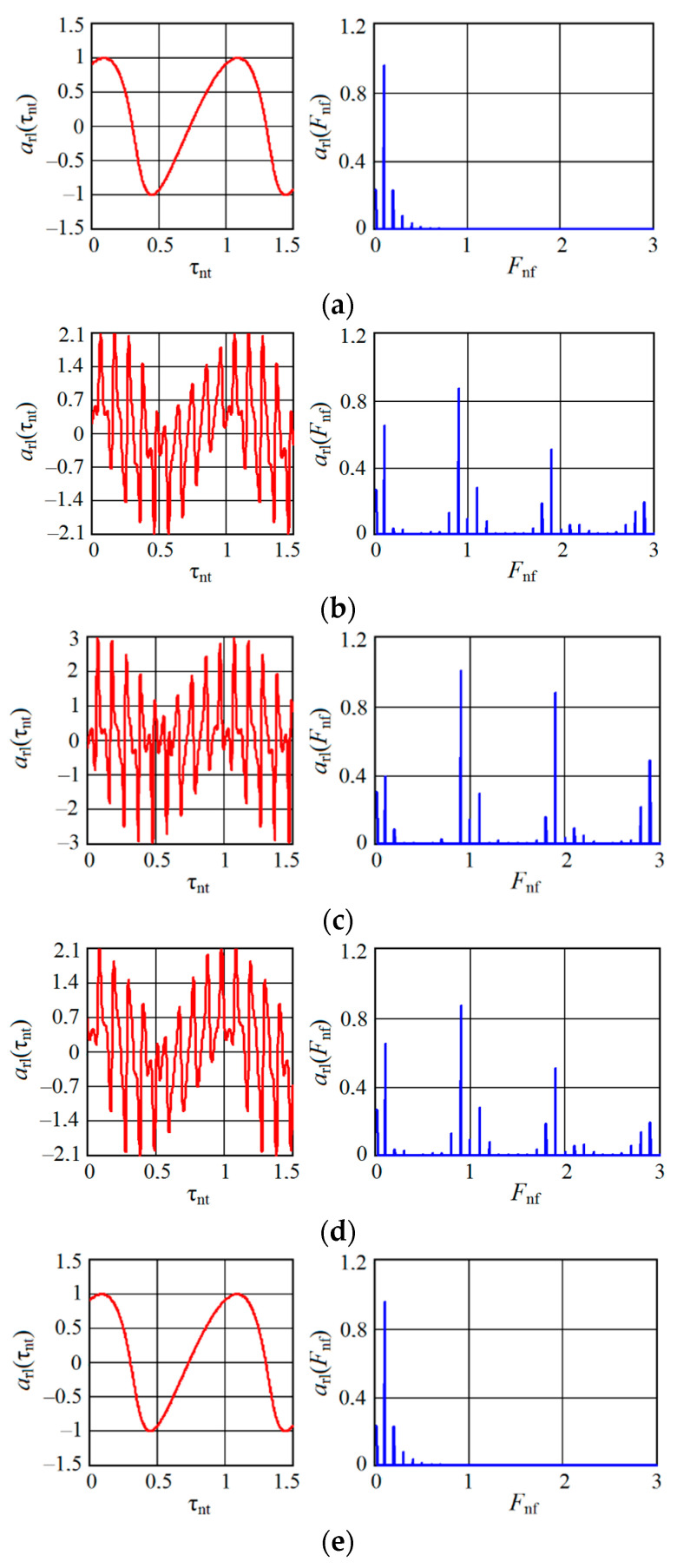
Figure **8.** Time $a_{\mathrm{rl}}(\tau_{\mathrm{nt}})$ and spectral $a_{\mathrm{rl}}(F_{\mathrm{nf}})$ diagrams of the autodyne signals calculated for the oscillator with AM in the presence of FM ($k_{\mathrm{FM}}=1$ ) and different values $r_{\mathrm{nd}}$: (**a**) $r_{\mathrm{nd}}=0$; (**b**) $r_{\mathrm{nd}}=0.25$; (**c**) $r_{\mathrm{nd}}=0.5$; (**d**) $r_{\mathrm{nd}}=0.75$; (**e**) $r_{\mathrm{nd}}=1$.

**Figure 9 sensors-20-07077-f009:**
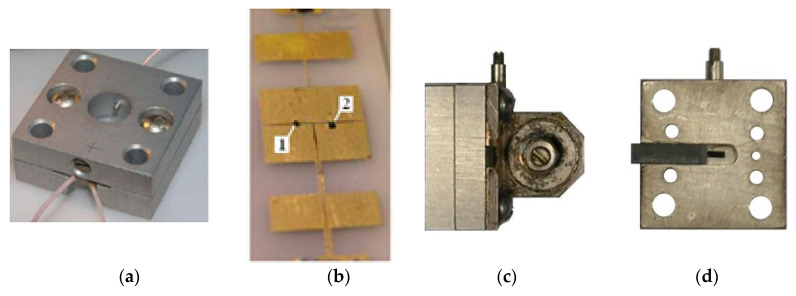
(**a**) External view of the Tigel-08M module, (**b**) topology of the diode insertion, (**c**) the oscillator with the stabilizing resonator and (**d**) the element of the resistor coupling with the resonator.

**Figure 10 sensors-20-07077-f010:**
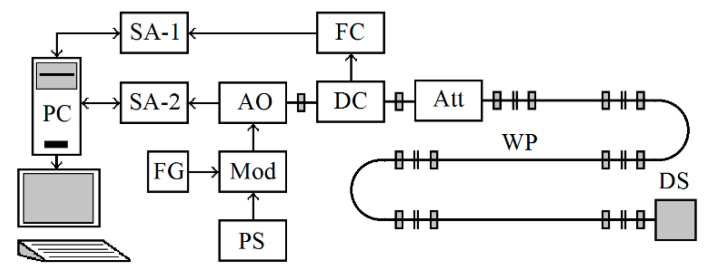
Block diagram of the experimental setup.

**Figure 11 sensors-20-07077-f011:**
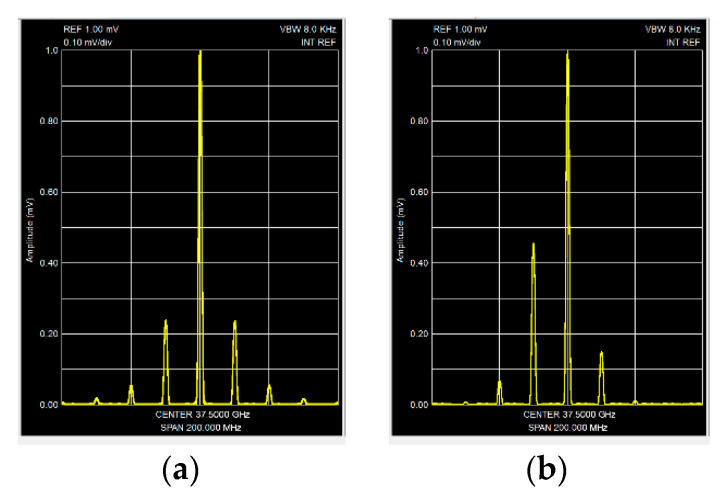
Spectrums of microwave oscillations of the output of the stabilized (**a**) and usual (**b**) (non-stabilized) oscillators.

**Figure 12 sensors-20-07077-f012:**
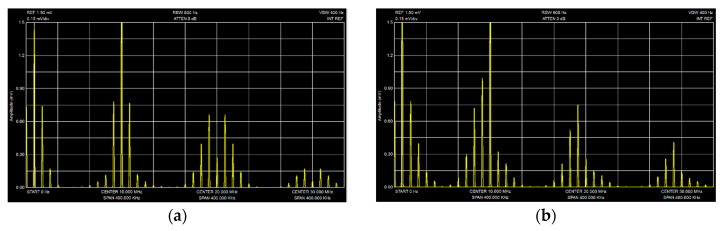
Spectrums of the autodyne signal obtained for the stabilized oscillator in the frequency on the Gunn diode with AM (**a**) and for usual (non-stabilized) oscillator, in which the attended FM is present (**b**).

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
