# Peer review of "Autodyne Sensor Signals with Amplitude-Frequency Modulation of Radiation"

_sensors, 2020, doi:10.3390/s20247077_

Round 1

Reviewer 1 Report

Fig 10 about the experimental setup is quite good, but an actual photo of the setup will be much better.

For the doppler simulator, authors refer to a paper. More details needed.

Did authors consider varying WP? Although experimental results section is not short, more details seem to be necessary. Step by step details about what has been done, with actual photos of the equipment used. 

For someone who is only interesting in applications but not the theory, reading the experimental results section should be enough or almost enough.

Author Response

Dear Reviewer!

We have revised spelling in our manuscript. All corrections are show/highlighted in manuscript.

In our future works we will take into account your recommendation about photo of the experimental setup, but unfortunately now another experimental setup is in use in our laboratory and we can not rebuild previous one, now we are working in another frequency band. We hope that Fig 10 is quite informative for readers.

We are also glad to see that doppler simulator is interesting for you, but it is very simple device which is detailed in “Noskov, V.Ya., Ignatkov, K.A., Shaydurov, K.D. (2019) Frequency deviation of injection-locked microwave autodynes. Radioengineering, 28(4), pp. 721–728. DOI: 10.13164/re.2019.0721.” We are not sure that we should copy the material from our previous article.

Earlier in our Russian language article “Noskov, V.Ya., Smolskiy, S.M. (2011) Autodyne effect in oscillators with amplitude modulation. Radiotechnika. (2), pp. 21–36. (In Russian) http://radiotec.ru/article/8545” (not free access) we have already considered the effect of waveguide length varying. The results is quite interesting in our opinion and we would be glad publish them separately in English to make them global accessible.

Best regards, Authors.

Reviewer 2 Report

This paper proposes a mathematical model of an autodyne oscillator with AFM radiation has been developed, which
provides the ability to calculate the CAS, shape and spectrum of the autodyne signal for the general
case of an arbitrary ratio of the delay time of the radiation reflected from the target and the period of
the modulating function. The paper has the main impact into radar systems, particularly the short-range radar systems which nowadays are part of autonomous driving industrial challenge.
Therefore, the contribution of the paper is according to the industrial trend.

The experiments were performed by using a Tigel-08M Ka-band oscillator module.

Some English checks should be done:

- Aytodyne right in the title should be changed to Autodyne;
- Spectrumsofmicrowaveoscillationsof -> no spaces?
- theautodynesignalobtainedforthe stabilizedoscillatorinthefrequency -> again, no spaces? Please check the text.

Author Response

Dear Reviewer!

We have revised spelling in our manuscript. All corrections are show/highlighted in manuscript.

Best regards, Authors.

Reviewer 3 Report

The paper is well structured, presenting relevant results and conclusions for microwave and millemeter-wave SRAS. The  mathematical model of the autodyne oscillator itself is differentiated, which consider combined amplitude and frequency modulation. Measured results demonstrate the validity of the developed model. 

Author Response

(The authors gave the same response as above.)
